# Testing the practical utility of implicit measures of beliefs for predicting drunk driving

**Femke Cathelyn***, **Pieter Van Dessel, Jan De Houwer**

Department of Experimental Clinical and Health Psychology, Ghent University, Ghent, Belgium

* Femke.Cathelyn@UGent.be

**Data Availability Statement:** All (anonymized) data files for Study 1 and Study 2 are publicly available on the Open Science Framework at https://osf.io/97jf3/ and https://osf.io/vfygs/, respectively.

## Abstract

Despite the potential benefits of implicit measures over self-report measures, they are rarely used in real-world contexts to predict behavior. Two potential reasons are that (a) traditional implicit measures typically show low predictive validity and (b) the practical utility of implicit measures has hardly been investigated. The current studies test the practical utility of a new generation of implicit measures for predicting drunk driving. Study 1 ($N$ = 290) examined whether an implicit measure of beliefs about past drunk driving (i.e., the Past Driving Under the Influence Implicit Association Test; P-DUI-IAT) retrospectively predicts drunk driving in driving school students, a population for which this measure could have applied value. Study 1 also explored whether P-DUI-IAT scores prospectively predicted drunk driving over six months. Due to the low number of offenders, however, Study 1 had low statistical power to test this latter question. In Study 2 ($N$ = 228), we therefore examined the utility of the P-DUI-IAT and a new variant of this test (i.e., the Acceptability of Driving Under the Influence Implicit Association Test; A-DUI-IAT) to prospectively predict drunk driving in an online sample with a high number of offenders. Results from Study 1 show that the P-DUI-IAT predicts self-rated past drunk driving behavior in driving school students (ORs = 3.11–6.12, $p$s < .043, 95% CIs = [1.11, 37.69]). Results from Study 1 do not show evidence for utility of the P-DUI-IAT to prospectively predict self-rated drunk driving. Results from Study 2, on the other hand, show strong evidence for the utility of both implicit measures to prospectively predict self-rated drunk driving (ORs = 3.80–5.82, $p$s < .002, 95% CIs = [1.72, 14.47]). Although further applied research is necessary, the current results could provide a first step towards the application of implicit measures in real-world contexts.

## Introduction

Over the past 25 years, scholars have tested the predictive utility of implicit measures for several behavioral outcomes, such as political preferences [1], consumer behavior [2], deviant behavior [3], and racially biased behavior [4, 5]. Studies have shown that responding on some implicit measurement tasks, such as the Implicit Association Test (IAT; [6]), is less controllable than responding on self-report measurement tasks, such as questionnaires (e.g., [7–9]).

**Funding:** This work was supported by Special Research Fund, Ghent University grant BOF16/MET_V/002 to JDH and the Scientific Research Foundation Flanders grant FWO19/PDS/041 to PVD. The funders had no role in study design, data collection and analysis, decision to publish, or preparation of the manuscript. There was no additional external funding received for these studies.

**Competing interests:** The authors have declared that no competing interests exist.

Therefore, scholars deem implicit measures most useful for predicting behavior that is socially sensitive in nature, that is, behavior that individuals might not want to deliberately report on [10].

In an IAT designed to assess racial bias, for instance, participants are instructed to categorize stimuli (such as words or pictures) as fast as possible using two keyboard keys. In a first critical block, participants use one response key to categorize Black-related and negatively valenced stimuli, and one response key to categorize White-related and positively valenced stimuli. In a second critical block, Black-related and positively valenced stimuli share the same response key, whereas White-related and negatively valenced stimuli share the other response key. When participants respond faster in the first critical block than in the second critical block, it is assumed that they have an implicit pro-White bias.

Recently, implicit measures have been applied in the domain of traffic safety research (see [11] for a recent review). Because of their benefits over self-report measures, implicit measures seem to hold promise for application in real-world contexts within this domain [12, 13]. For instance, in certain countries, individuals are required to take a refresher course in a driving school a couple of months after obtaining their driver's license. In such a context, the application of implicit measures could be useful for predicting risky driving behavior, such as driving under the influence (DUI). In this context, asking people to self-report their DUI may not yield good results (because offenders may be dishonest to avoid negative consequences). Instead of (or in addition to) self-report measures, implicit measures could be used to detect which individuals are likely to drink and drive (again). Consequently, these individuals could be provided with interventions to prevent (further) offenses.

Importantly, however, meta-analyses so far have provided little evidence for the predictive utility of implicit measures (e.g., [4, 14]). One possible reason for this finding is that traditional implicit measures do not sufficiently specify how concepts of interest are related [15]. For example, in an IAT designed to assess attitudes towards drunk driving, a participant might reveal faster responding in blocks in which words such as "bad" and words such as "drunk driving" share the same response key either because they personally believe that drunk driving *is* bad, or because they believe that drunk driving *is typically considered by others* as bad. Traditional implicit measures such as the IAT are not able to distinguish between these beliefs, even though these beliefs could have different behavioral effects. Considering this observation, it might not come as a surprise that studies have shown little evidence for the predictive utility of traditional implicit measures for drunk driving [16, 17]. Recently, researchers have started to develop a new generation of implicit measures aimed at capturing specific beliefs. These measures employ more complex propositional stimuli that specify the relationship between concepts (e.g., "drunk driving is bad") and probe truth evaluation of these stimuli. As such, these implicit measures allow for probing beliefs. Implicit measures of beliefs seem to hold promise for predicting different types of behavior [18–21]. Moreover, initial evidence suggests that these measures outperform traditional implicit measures when predicting behavior [5, 22].

A second issue in implicit measures research is that scholars rarely examine the *practical* utility of implicit measures. First, the predictive utility of implicit measures is hardly ever tested in a setting or population for which scholars consider implicit measures to have applied value. For instance, in a recent meta-analysis on the predictive utility of implicit measures for racial bias [5], only 23 out of 225 studies were conducted in a real-world setting, whereas the remaining studies were conducted in a lab setting, with the majority of studies testing undergraduate students. Such methodological limitations jeopardize the ecological validity of findings. In the context of traffic safety, for instance, scholars have argued for the application of implicit measures in driving schools [12, 13], however, no studies thus far have tested the predictive utility of implicit measures in these populations.

Second, for a prediction measure to have applied value, its utility to *prospectively* predict behavior should be tested. Nevertheless, implicit measures have rarely been put to this test (for an exception, see studies on self-harm behavior [23]). For instance, in a recent review, Schmidt, Banse, and Imhoff [24] discuss several studies demonstrating the IAT's utility to retrospectively predict sexual deviant preferences, but indicate that "data on predictive validity, the most relevant piece of the puzzle for applied purposes, are still missing" (p. 192). Similarly, in the domain of traffic research, to the best of our knowledge, no studies have yet investigated the utility of implicit measures to predict risky driving behaviors over time (see [11] for a recent review).

The current studies aimed to address these limitations and test the practical utility of implicit measures of beliefs for predicting drunk driving. In previous studies [25], we conducted an initial validation test of an implicit measure of beliefs for detecting drunk driving: the past driving under the influence IAT (P-DUI-IAT). The P-DUI-IAT follows the same procedure as a traditional IAT, with the exception that its stimuli contain full sentences instead of single words (also see [26]). In the P-DUI-IAT, participants are asked to categorize sentences regarding past or non-past drunk driving (e.g., "drunk driving is something I have done" or "drunk driving is something I have not done") together with sentences that are inherently true (e.g., "I'm doing a computer task") or false (e.g., "I'm playing football). The extent to which a participant responds faster to the combination of inherently true sentences and sentences regarding past drunk driving is thought to provide an index of the extent to which that participant automatically endorses the belief that he or she has driven drunk in the past. Results from our previous studies showed that P-DUI-IAT scores were higher for participants who indicated to have driven drunk in the past than for participants who reported to never have driven drunk. Results also showed that P-DUI-IAT scores predicted self-rated future likelihood of drunk driving.

The current paper reports two studies. The aim of Study 1 was to validate the P-DUI-IAT in a sample of driving school students who took the obligated refresher course after obtaining their driver's license. Unlike previous studies with the P-DUI-IAT that were conducted in online samples [25], Study 1 thus tested the P-DUI-IAT in an ecologically valid situation. Study 1 also explored whether P-DUI-IAT scores prospectively predicted drunk driving over a period of six months. However, because only few participants recruited within the ecological setting of the driving schools reported DUI offenses, our analyses only attained low statistical power to detect effects at follow-up.

A first aim of Study 2 was to systematically test the utility of the P-DUI-IAT to prospectively predict drunk driving (over a period of 30 days) using sample sizes that allowed higher statistical power to detect effects. Therefore, in Study 2, we used a platform for online participant recruitment which allowed us to run a prescreening study with the aim of recruiting a larger number of participants who would likely drink and drive between baseline and follow-up (i.e., participants who had recently driven drunk).

In light of the lack of a prospective predictive validity effect of the P-DUI-IAT in Study 1, a second aim of Study 2 was to test the prospective predictive validity of a newly developed implicit measure: the acceptability of driving under the influence IAT (A-DUI-IAT). Notably, the P-DUI-IAT refers to past behavior and could therefore only be used to predict past behavior and the probability of re-occurrence of drunk driving. It cannot, however, be used to predict the first onset of drunk driving behavior (e.g., in driving school students who have not obtained their driver's license yet). As a result, its application options would be limited. The A-DUI-IAT, on the other hand, probes beliefs about the personal acceptance of drunk driving (i.e., endorsement of sentences such as "drunk driving is acceptable to me") and would therefore be better suited for predicting the onset of DUI.

A third and final aim of Study 2 was to test whether we could replicate previous findings (i.e., of [25] and Study 1) regarding validity of the P-DUI-IAT or, in other words, to test its utility to distinguish between past drunk driving offenders and non-offenders. Please note that analyses regarding the retrospective predictive utility of the A-DUI-IAT are less relevant for validating this measure (because the A-DUI-IAT does not probe beliefs regarding past drunk driving) and are therefore presented in the Supplementary Information (see S1 Appendix) of this paper.

## Method

All anonymized data files, study and analytic scripts of Study 1 and Study 2 are publicly available on the Open Science Framework (see https://osf.io/97jf3/ and https://osf.io/vfygs/, respectively). The study design, sampling, and analysis plan of both studies were preregistered (see https://osf.io/8r9j7/ and https://osf.io/anzqw/ for the preregistrations of Study 1 and Study 2, respectively). The ethical committee of the Faculty of Psychology and Educational Sciences at Ghent University approved both studies. The study procedures were carried out in accordance with the Declaration of Helsinki. All subjects were informed about the study and provided informed consent. All participants were over the age of 18. Given that the studies were conducted online, written consent could not be obtained. Instead, participants were asked to (virtually) check one of two boxes: "Yes, I consent to participate in this study" or "No, I do not consent to participate in this study". If subjects checked the latter option, the study was automatically terminated. These responses were timestamped and stored alongside the subjects' email addresses (Study 1) or Prolific IDs (Study 2).

### Participants

Five Belgian driving schools invited native Dutch-speaking students who had recently taken the refresher course to participate in Study 1. The invitation email included information about the study, inclusion criteria (i.e., Dutch as native language), and a link to a website that hosted the study online. After completing the study, participants received a five euro gift voucher. Large enough between-group differences are required for an adequate test of the practical value of the P-DUI-IAT. Based on this requisite and effect sizes observed in previous studies [25], we planned to recruit at least 290 participants, including at least 26 participants who had driven drunk and 264 who had not, because these samples sizes would allow 90% power to detect a medium effect size ($d$ = .60, alpha = .05, one-tailed) in a $t$-test comparing IAT scores between these groups.

A total of 457 participants started Study 1. In line with our preregistered plan, the data of participants were excluded who did not provide complete data ($n$ = 41) or met the exclusion criteria of the IAT D4-scoring procedure ($n$ = 118; i.e., response latencies less than 300 ms on 10% or more of the critical trials, error rates above 30% for all of the critical blocks, and/or error rates above 40% for any of the critical blocks). Additionally, the data were excluded of eight participants who indicated to have driven drunk in the past month but not since obtaining their driver's license or who indicated to not have a driver's license. The final sample size consisted of 290 participants. This sample consisted of 246 participants who had not driven drunk since obtaining their driver's license and 44 participants who had. These sample sizes provided 98% power to detect a medium effect size ($d$ = .60, alpha = .05, one-tailed). Sample characteristics are presented in Table 1.

Six months after completing the baseline measures, participants with complete data and a correct identification code ($n$ = 285) were asked about drunk driving behavior during the six-month period. The question was answered by 141 participants.

**Table 1. Sample characteristics Study 1 per group.**

|  | Drunk driving since driver's license (*n* = 44) | No drunk driving (*n* = 246) |
|---|---|---|
| Age, *M (SD)* | 21.73 (6.14) | 20.65 (3.21) |
| Gender |  |  |
| % male (*n*) | 52.30% (23) | 38.60% (95) |
| % female (*n*) | 45.50% (20) | 59.30% (146) |
| % other (*n*) | 2.30% (1) | 2% (5) |
| Number of months in possession of driver's license, *M (SD)* | 19.87 (47.90) | 12.64 (10.12) |
| Weekly mileage, *M (SD)* | 103.11 (112.01) | 75.33 (114.73) |
| Units of alcohol per week, *M (SD)* | 6.14 (7.12) | 2.18 (3.94) |

In Study 2, native English-speaking participants were recruited via Prolific Academic (an online recruitment platform). We first ran a short prescreening study to recruit a larger number of participants who would likely driving and drive between baseline and follow-up (i.e. participants who had recently driven drunk). Participants who owned a valid driver's license, drove their car at least once per week, drank more than one unit of alcohol per week, had the UK nationality, and whose first language was English, were invited to participate in the prescreening study. Participants who indicated during the prescreening study to either (a) have no history of drunk driving (*n* = 240), (b) having driven drunk in the past year (*n* = 120), and (c) having driven drunk in the past month (*n* = 120) were invited to participate in the main study.

We planned to have a sufficient number of participants to have 90% power to detect a medium effect size (*d* = 0.70, alpha = .05, one-tailed) in the between-groups *t*-test comparing IAT scores between drunk driving groups at follow-up. We estimated that 480 participants would allow for sufficient power, taking into account possible drop out between baseline and follow-up (estimated at 75%) and taking into account that we would need a sufficient number of participants to have engaged in DUI behavior in the 30-day period (estimated at 35%).

From the 480 invited participants, 312 started the main study. The data were excluded of 46 participants who met the exclusion criteria of the IAT D4-scoring procedure for both IATs (*n* = 19) or did not provide complete data (*n* = 27). The final sample size consisted of 266 participants. For the follow-up analyses, the final sample size consisted of 228 participants. This sample included 65 participants who had driven drunk between baseline and follow-up and 163 participants who had not. These final sample sizes provided 99% power to detect medium effect sizes (*d* = 0.70, alpha = .05, one-tailed) in the between-groups *t*-test comparing IAT scores for drunk driving at follow-up. Participants received a small monetary reward upon completing the prescreening study (£0.13), part 1 of the main study (£1.25), and part 2 of the main study (£1.50). The sample characteristics are presented in Table 2.

## Materials

For Study 2, we adopted the (English) materials from our previous studies [25]. For Study 1, all materials were translated to Dutch using the back translation method.

**Implicit measures of drunk driving.** The P-DUI-IAT followed the same procedure as in our previous studies [25]. Participants were instructed to categorize statements as fast as possible using two keys on the keyboard (the "E" and "I" keys). On each trial, a statement appeared in the middle of the screen. If the response was correct, the stimulus disappeared, and the next stimulus was presented 400ms later. If the response was incorrect, a red cross replaced the

**Table 2. Sample characteristics Study 2 per group.**

|  | Drunk driving past year (*n* = 141) | No past drunk driving (*n* = 125) | Prospective drunk driving (*n* = 65) | No prospective drunk driving (*n* = 163) |
|---|---|---|---|---|
| Age, *M (SD)* | 35.77 (11.64) | 40.81 (13.76) | 37.26 (12.58) | 38.74 (13.11) |
| Gender |  |  |  |  |
| % male (*n*) | 57.40% (81) | 32% (40) | 61.50% (40) | 41.7% (68) |
| % female (*n*) | 42.60% (60) | 68% (85) | 38.50% (25) | 58.30% (95) |
| Years of driving experience, *M (SD)* | 19.64 (37.32) | 20.92 (14.62) | 19.03 (12.29) | 21.76 (35.54) |
| Weekly mileage, *M (SD)* | 149.08 (203.58) | 103.84 (117.71) | 149.78 (120.11) | 121.91 (187.55) |
| Units of alcohol per week, *M (SD)* | 13.73 (16.23) | 6.04 (11.43) | 16.58 (16.35) | 7.44 (13.23) |

stimulus for 200ms, and the next stimulus appeared 400ms after the red cross appeared. There were two types of statements: statements regarding past drunk driving (e.g., "I have driven while being drunk" or "I have always driven while sober") and statements that were logically true or false (e.g., "I'm doing a computer task" or "I'm climbing a mountain"). All of the items for the P-DUI-IAT are listed in the Supporting Information files of this paper (see S1 Table). Labels for the past drunk driving categories (i.e., I HAVE DRIVEN DRUNK BEFORE and I HAVE NEVER DRIVEN DRUNK) and true/false categories (i.e., TRUE and FALSE) were presented in the top left and right corners to aid categorization.

In the first block, participants practiced categorizing statements regarding (not) past drunk driving, and in the second block, participants practiced categorizing true/false statements. For past drunk driving and inherently true statements, participants pressed the E-key, and for not drunk driving and inherently false statements, participants pressed the I-Key. Each block consisted of 24 trials. In the third block, participants categorized statements from all four categories using the key assignment that they practiced in the previous blocks, for 48 trials. Next, participants practiced categorizing statements regarding (not) past drunk driving, but this time, with the response key assignment reversed (i.e., E-key for not drunk driving statements and I-key for past drunk driving statements). This block consisted of 24 trials. Finally, participants completed 48 critical trials in which they categorized statements from all four categories using the new response key assignment.

The A-DUI-IAT followed the same procedure as the P-DUI-IAT, with the exception that statements regarding past drunk driving were replaced with statements regarding the personal acceptance of drunk driving, such as "Driving after drinking alcohol is acceptable to me" and "I'm opposed to driving after drinking alcohol" The category labels and all of the items for the A-DUI-IAT are listed in the Supporting Information files of this paper (see S2 Table).

Scores for the P-DUI-IAT (the Spearman-Brown corrected split-half reliability equaled .59 in Study 1 and .72 in Study 2) and A-DUI-IAT (Spearman-Brown corrected split-half reliability = .68) were calculated using the D4 scoring algorithm [27]. Reaction times on trials of the first critical block were subtracted from reaction times on trials of the second critical block, such that higher scores indicated faster responding in critical blocks in which statements indicating past DUI behavior or acceptance of DUI behavior and statements that were logically true shared the same response key.

**Self-report measures of drunk driving.** Past and prospective drunk driving was assessed by asking participants how many times they had driven their car when they might have exceeded the legal limit for drinking and driving (a) since obtaining their driver's license (Study 1) or in the past year (Study2), (b) in the past month, and (c) between baseline and follow-up. In Study 1, participants could answer these questions by inserting any number. In

Study 2, participants were asked to indicate frequency of drunk driving on a scale (ranging from *0 times* to *10+ times*). Self-rated future likelihood of drunk driving was measured by asking participants how likely they would be to drink and drive (again) in the future. Responses were given on a Likert scale ranging from one (*very unlikely*) to five (*very likely*).

**Measures of risk factors.** To measure alcohol consumption, we asked participants how many units of alcohol they drink on average per week. Perceived behavioral control (PBC) was measured using a subscale of a questionnaire developed by Marcil and colleagues [28]. Before answering the questions, participants were instructed to imagine that they drove their car to a party where they drank alcohol but were uncertain whether their blood alcohol level exceeded the legal limit when they had to return home. This subscale consisted of five questions (e.g., "For me, driving my car after drinking alcohol at the party is. . ."). Questions were answered on a bipolar scale ranging from -3 (e.g., *easy*) to +3 (e.g., *difficult*). Scores for each question were averaged to obtain the total score (Cronbach's Alpha = .93).

## Procedure

In Study 1, participants first answered demographical questions and questions regarding their car and alcohol use. Next, participants completed the P-DUI-IAT. Before completing the scales and questions regarding drunk driving, participants were reminded about the anonymous nature of the study. Participants first answered questions regarding past and future likelihood of drunk driving and then completed the PBC scale. Six months after baseline measures, participants were asked about drunk driving behavior during the follow-up period.

The procedure of Study 2 was identical to the procedure of Study 1, with the exception that participants completed a second IAT at the end of the study. The order of IATs was counterbalanced between participants. One month after baseline measures, participants were invited to participate in the second part of the study. At follow-up, participants were asked whether they had driven drunk during the one-month period.

## Data analysis

To examine the utility of the implicit measures to discriminate between participants with and without a history of drunk driving, we used two-sample *t*-tests. To examine how well the implicit measures discriminate between these groups, we conducted receiver-operating-characteristic (ROC) analyses. In our previous studies [25], we tested different cut-off points of the P-DUI-IAT to maximize either sensitivity (true positive rate) or specificity (true negative rate). We examined whether these cut-off points remained meaningful in the current sample. For the A-DUI-IAT, we established new cut-off points to maximize sensitivity or specificity while retaining fair specificity and sensitivity, respectively.

To examine the utility of the implicit measures to independently predict past and future likelihood of drunk driving, we performed logistic regression analyses. To examine the utility of the implicit measures to predict past drunk driving and future likelihood of drunk driving above and beyond known risk factors (i.e., PBC, average units of alcohol per week, age, and gender for the prediction of past drunk driving, as well as frequency of past drunk driving for the prediction of future likelihood of drunk driving), we used hierarchical regression analyses. For these analyses, significant risk factors were added in the first step and IAT scores were entered in the second step.

Finally, to examine whether the implicit measures were capable of prospectively predicting drunk driving, we conducted the same analyses as described above. As indicated in the preregistration of Study 1, if we recruited fewer than 20 participants who had driven drunk between baseline and follow-up we would consider analyses regarding the utility of the P-DUI-IAT to

**Table 3. Number of participants per DUI group Study 1.**

| Group | n |
|---|---|
| Past DUI group | 246 |
| Past month DUI group | 12 |
| No history of DUI group | 44 |
| Low future likelihood DUI group | 261 |
| High future likelihood DUI group | 29 |
| Prospective DUI group | 17 |
| No prospective DUI group | 124 |

*Note*. DUI = driving under the influence.

prospectively predict drunk driving as exploratory rather than confirmatory analyses (given the low statistical power).

For the analyses regarding the prediction of past DUI behavior, participants were grouped based on the questions regarding past DUI frequency (e.g., participants who indicated to have driven drunk zero times in the past were assigned to the no drunk driving group). For the analyses regarding the prediction of future likelihood of drunk driving, participants were assigned to the low likelihood group if they had a score of one or two on the future likelihood scale and to the high future likelihood group if they had a score between three and five. Participants who indicated to have driven drunk more than zero times between baseline and follow-up were assigned to the prospective DUI group and participants who indicated to have driven drunk zero times between baseline and follow-up were assigned to the no prospective DUI group (regardless of drunk driving history as indicated at baseline). Table 3 describes the number of participants per DUI group for Study 1. Table 4 describes the number of participants per DUI group for Study 2. Note that not all participants had IAT scores for both IAT types given that participants were only excluded if they met the exclusion criteria of the IAT D4-scoring procedure for *both* IATs, and thus, the number of participants per IAT type slightly differed (see Table 4).

## Deviations from preregistration

There were four deviations from the preregistered plan for Study 1. First, besides excluding the data of participants based on our preregistered exclusion criteria (i.e., incomplete data and

**Table 4. Number of participants per DUI group and IAT type Study 2.**

| Group | n (P-DUI-IAT scores) | n (A-DUI-IAT scores) |
|---|---|---|
| Past DUI group | 132 | 136 |
| Past month DUI group | 84 | 88 |
| No history of DUI group | 119 | 116 |
| Low future likelihood DUI group | 175 | 172 |
| High future likelihood DUI group | 76 | 80 |
| Prospective DUI group | 61 | 62 |
| No prospective DUI group | 154 | 153 |

*Note*. P-DUI-IAT = past driving under the influence implicit association test; A-DUI-IAT = acceptability of driving under the influence implicit association test. For the sake of completeness, this table also reports the number of participants with A-DUI-IAT scores for the past drunk driving groups and future likelihood groups. However, in the current paper, we only report analyses regarding the prospective predictive utility of the A-DUI-IAT and thus only compared the prospective drunk driving groups.

exclusion criteria of the IAT D4-scoring procedure), we also excluded the data of (a) participants who indicated to not have a driver's license (because these participants were either no driving school students or they were not paying attention during the study) and (b) participants who indicated to have driven drunk in the past month but not since obtaining their driver's license (because we could not determine to which group these participants should be assigned). The patterns of results was similar when excluding the data of these participants. Second, we preregistered that we would assess the utility of the P-DUI-IAT to independently predict prospective drunk driving (using logistic regression), but we forgot to preregister that we would also assess this for the past drunk driving outcome variables. Third, we preregistered that we would conduct hierarchical *linear* regression analyses to examine the predictive validity of the P-DUI-IAT for self-rated future likelihood of drunk driving (rated on a Likert scale). However, given that the majority of participants scored zero on this question, variability for the future likelihood variable was low, and thus, covariance with the independent variable would be artificially lowered [29]. Therefore, it is more appropriate to use logistic rather than linear regression analyses to assess the relationship between future likelihood of drunk driving and IAT scores. Finally, we also compared IAT scores between the prospective drunk driving group and non-prospective drunk driving group using a Bayesian *t*-test, which allows estimating the amount of evidence for the null hypothesis.

There were no deviations from the preregistered plan for Study 2, with the exception that we did not only recruit participants from the United Kingdom, but also participants from the United States. This was done because, during the pre-screening study, we were not able to recruit the planned number of participants that we wanted to invite for the main study. Subsequently, we used British- and American-English versions of the IATs (i.e., for the American-English version of the IATs we replaced "drink" driving with "drunk" driving). Nationality did not moderate the effects, $\beta$s < 0.72, $p$s > .38.

## Results

### Validation of the P-DUI-IAT in a sample of driving school students

Results from Study 1 showed that P-DUI-IAT scores were significantly lower for participants without a history of drunk driving ($M$ = -0.04, $SD$ = 0.35) than for participants who had driven drunk since obtaining their driver's license ($M$ = 0.11, $SD$ = 0.45), $t$(52.59) = 2.14, $d$ = 0.42, $p$ = .018 and participants who had driven drunk in the past month ($M$ = 0.17, $SD$ = 0.35), $t$(12.10) = 2.07, $d$ = 0.61, $p$ = .029.

The Area Under the Curve (AUC) was .59 (95% CI = .48-.69) for drunk driving since obtaining one's driver's license and .66 (95% CI = .50-.82) for past month drunk driving. The previously determined IAT cut-off score to maximize sensitivity and retain fair specificity (-0.08) produced 55% sensitivity and 46% specificity to detect drunk driving since obtaining one's driver's license, and 75% sensitivity and 46% specificity to detect past month drunk driving. The previously determined IAT cut-off score to maximize specificity and retain fair sensitivity (0.41) produced 91% specificity and 27% sensitivity for the detection of drunk driving since obtaining one's driver's license, and 91% specificity and 33% sensitivity for the detection of past month drunk driving.

Higher P-DUI-IAT scores were significantly associated with drunk driving since obtaining one's driver's license, OR = 3.11, 95% CI = [1.29, 7.70], $p$ = .012, past month drunk driving, OR = 6.12, 95% CI = [1.11, 37.69], $p$ = .042, and self-rated future likelihood of drunk driving, OR = 3.28, 95% CI = [1.15, 9.56], $p$ = .029. Significant risk factors of drunk driving for each outcome (see S3 Table) were statistically controlled for in the hierarchical regression analyses. Results revealed that P-DUI-IAT scores did not show incremental validity for the prediction of any of the outcome measures, $\chi^2$s < 3.22, $p$s > .06.

## Exploring the utility of the P-DUI-IAT to prospectively predict drunk driving in a sample of driving school students

Results from Study 1 showed that P-DUI-IAT scores were not different for the group that had driven drunk between baseline and follow-up ($M$ = 0.01, $SD$ = 0.48) than for the group that had not ($M$ = -0.03, $SD$ = 0.34), $t$(18.32) = -0.33, $d$ = 0.11, $p$ = .627. Bayesian $t$-test analyses revealed a Bayes factor of 0.28, indicating moderate evidence for the null hypothesis. The AUC for prospective drunk driving was .53, which is around chance level (.50). The -0.08 cut-off score produced 59% sensitivity and 45% specificity to detect drunk driving during follow-up. The 0.41 cut-off score produced 94% specificity and 18% sensitivity to detect drunk driving during follow-up.

Higher P-DUI-IAT scores were not significantly associated with drunk driving during the six-month follow-up period, OR = 1.36, 95% CI = [.33, 5.56], $p$ = .67 and P-DUI-IAT scores did not predict this outcome above and beyond the significant known risk factor, $\chi^2$(1) = .01, $p$ = .925. Please note that only one risk factor (frequency of drunk driving since obtaining one's driver's license) was significant in the prediction of prospective drunk driving (see S3 Table).

## Testing the utility of the P-DUI-IAT and A-DUI-IAT to prospectively predict drunk driving in an online sample

Results from Study 2 showed that there was a significant difference in P-DUI-IAT scores between participants who had driven drunk between baseline and follow-up ($M$ = 0.28, $SD$ = 0.35) and participants who did not ($M$ = 0.08, $SD$ = 0.40), $t$(123.65) = 3.57, $d$ = 0.51, $p <$ .001. Analyses also revealed a significant difference in A-DUI-IAT scores between these two groups ($M$ = 0.37, $SD$ = 0.37 for the prospective drunk driving group and $M$ = 0.14, $SD$ = 0.36 for the prospective non-drunk driving group), $t$(111.51) = 4.25, $d$ = 0.64, $p <$ .001.

The overall ability of the P-DUI-IAT and A-DUI-IAT to correctly classify participants as prospective (non-) drunk drivers (i.e., the AUC) was .65 (95% CI = .57-.73) and .66 (95% CI = .58-.74), respectively. Assigning participants to the prospective drunk driving groups based on P-DUI-IAT scores using the -0.08 threshold produced 85% sensitivity and 34% specificity, while using the 0.41 threshold produced 79% specificity and 30% sensitivity. Using -0.07 as a cut-off score for the A-DUI-IAT produced maximum sensitivity (89%) while retaining fair specificity (30%) for the detection of prospective drunk driving. Using 0.57 as a cut-off score for the A-DUI-IAT produced maximum specificity (87%) while retaining fair sensitivity (31%).

Higher P-DUI-IAT and A-DUI-IAT scores were significantly associated with drunk driving at follow-up, with an OR of 3.80 (95% CI = 1.72–8.86, $p$ = .001) for P-DUI-IAT scores and an OR of 5.82 (95% CI = 2.50–14.47, $p <$ .001) for A-DUI-IAT scores. To examine incremental validity of the IATs in the prediction of prospective drunk driving using hierarchical regression analyses, significant risk factors were entered in the first step (see S4 Table) and IAT scores were entered in the second step. Analyses showed that P-DUI-IAT scores did not predict prospective drunk driving above and beyond known risk factors, $\chi^2$ = 0.11, $p$ = .74. The difference between the model including significant risk factors and the model including significant risk factors and A-DUI-IAT scores, however, was marginally significant, $\chi^2$ = 3.92, $p$ = .048 (see Table 5).

## Testing the replicability of previous findings: Utility of the P-DUI-IAT to predict past DUI and future likelihood of DUI in an online sample

Results from Study 2 showed that P-DUI-IAT scores were significantly lower for participants without a history of drunk driving ($M$ = -0.01, $SD$ = 0.37) than for participants who had driven

**Table 5. Hierarchical logistic regression predicting prospective drunk driving (Study 2).**

| Variable | B | SE | Wald | OR (95% CI) | $\chi^2$ | $R^2$ |
|---|---|---|---|---|---|---|
| Step 1 | | | | | $\chi^2(5) = 120.68^{***}$ | 0.61 |
| Gender (male) | -0.05 | 0.48 | .01 | 0.96 (0.37, 2.44) | | |
| Units of alcohol | 0.02 | 0.02 | 0.99 | 1.02 (0.98, 1.04) | | |
| PBC | 0.34 | 0.14 | 5.50 | 1.40* (1.06, 1.87) | | |
| DUI past year | 0.55 | 0.13 | 16.90 | 1.73*** (1.35, 2.28) | | |
| DUI past month | 0.37 | 0.35 | 1.10 | 1.45 (0.78, 3.12) | | |
| Step 2 | | | | | | |
| A-DUI-IAT scores | 1.27 | 0.66 | 3.70 | 3.55 (1.01, 13.55) | $\chi^2(1) = 3.92$ | 0.63 |

*Note.* PBC = perceived behavioral control; A-DUI-IAT = acceptability of driving under the influence implicit association test; OR = odds ratio; CI = confidence interval.

* $p < .05$.

** $p < .01$.

*** $p < .001$.

drunk in the past year ($M = 0.27$, $SD = 0.36$), $t(245.64) = 6.03$, $d = 0.76$, $p < .001$, and participants who had driven drunk in the past month ($M = 0.30$, $SD = 0.37$), $t(179.06) = 2.07$, $d = 0.84$, $p < .001$.

The AUC was .70 (95% CI = .64-.77) for past year drunk driving and .72 (95% CI = .65-.80) for past month drunk driving, which is well above chance level (.50). The threshold to maximize sensitivity and retain fair specificity (-0.08 IAT score) produced 83% sensitivity and 42% specificity for detecting past year drunk driving and 87% sensitivity and 42% specificity for detecting past month drunk driving. The threshold to maximize specificity and retain fair sensitivity (-0.41 IAT score) produced 85% specificity and 30% sensitivity for detecting past year drunk driving and 85% specificity and 31% sensitivity to detect past month drunk driving.

Higher P-DUI-IAT scores were significantly associated with past year drunk driving, OR = 8.47, 95% CI = [3.97, 19.25], $p < .001$, past month drunk driving, OR = 10.55, 95% CI = [4.42, 27.64], $p < .001$, and self-rated future likelihood of drunk driving, OR = 4.94, 95% CI = [2.33, 11.10], $p < .001$. Significant risk factors of drunk driving for each outcome (see S4 Table) were statistically controlled for in the hierarchical regression analyses. Results revealed that P-DUI-IAT scores predicted past year drunk driving (see Table 6) and past month drunk driving (see Table 7) above and beyond known risk factors. The P-DUI-IAT did not show incremental validity for the prediction of future likelihood of drunk driving, $\chi^2 = 0.00$, $p = .983$.

**Table 6. Hierarchical logistic regression predicting past year drunk driving (Study 2).**

| Variable | B | SE | Wald | OR (95% CI) | $\chi^2$ | $R^2$ |
|---|---|---|---|---|---|---|
| Step 1 | | | | | $\chi^2(4) = 92.99^{***}$ | 0.41 |
| Gender (male) | 0.49 | 0.32 | 2.30 | 1.62 (0.87, 3.04) | | |
| Age | -0.04 | 0.01 | 9.10 | 0.96* (0.94, 0.99) | | |
| Units of alcohol | 0.05 | 0.02 | 7.80 | 1.05** (1.02, 1.09) | | |
| PBC | 0.64 | 0.11 | 36.20 | 1.90*** (1.56, 2.37) | | |
| Step 2 | | | | | $\chi^2(1) = 9.41^{**}$ | 0.45 |
| P-DUI-IAT scores | 1.33 | 0.45 | 8.90 | 3.77** (1.60, 9.27) | | |

*Note.* PBC = perceived behavioral control; P-DUI-IAT = past driving under the influence implicit association test; OR = odds ratio; CI = confidence interval.

* $p < .05$.

** $p < .01$.

*** $p < .001$.

**Table 7. Hierarchical logistic regression predicting past month drunk driving (Study 2).**

| Variable | B | SE | Wald | OR (95% CI) | $\chi^2$ | $R^2$ |
|---|---|---|---|---|---|---|
| Step 1 | | | | | $\chi^2(4) = 94.54^{***}$ | 0.50 |
| Gender (male) | 0.43 | 0.39 | 1.20 | 1.54 (0.71, 3.28) | | |
| Age | -0.04 | 0.02 | 6.00 | $0.96^*$ (0.93, 0.99) | | |
| Units of alcohol | 0.04 | 0.02 | 6.80 | $1.05^{**}$ (1.02, 1.09) | | |
| PBC | 0.76 | 0.13 | 38.70 | $2.14^{***}$ (1.71, 2.77) | | |
| Step 2 | | | | | | |
| P-DUI-IAT scores | 1.52 | 0.54 | 7.90 | $4.55^{**}$ (1.63, 13.72) | $\chi^2(1) = 8.57^{**}$ | 0.54 |

*Note.* PBC = perceived behavioral control; P-DUI-IAT = past driving under the influence implicit association test; OR = odds ratio; CI = confidence interval.

$^*$ $p < .05$.

$^{**}$ $p < .01$.

$^{***}$ $p < .001$.

## General discussion

In this paper, we report two studies testing the practical utility of implicit measures of beliefs for predicting drunk driving. Study 1 showed initial evidence for validation of the P-DUI-IAT in driving school students who took the refresher course, a population for which this measure could have applied value. Results of Study 2 showed initial evidence for the utility of the P-DUI-IAT and A-DUI-IAT to prospectively predict drunk driving in online samples and replicated findings from previous studies.

### Summary and interpretation of findings

In line with results from our previous studies [25], Study 1 showed that the P-DUI-IAT discriminated between driving school students with and without a history of drunk driving, and higher P-DUI-IAT scores were associated with self-reports of past drunk driving behavior and self-reports of future likelihood of drunk driving. The Results from Study 1 did not show evidence for the utility of the P-DUI-IAT to predict the outcome measures above and beyond known risk factors (as opposed to results from previous studies [25] and Study 2). It is also of note that the study including driving school students produced more modest group differences ($d = 0.42$) than studies including online samples ($d = 0.85$ in [25]; $d = 0.76$ in Study 2). A possible reason for these differences in findings is that driving school students were more motivated to hide drunk driving behavior than participants from online samples (see below for a further discussion). As such, it is possible that the P-DUI-IAT detected more cases than could be observed using the current data. A second plausible reason for the difference in effects sizes is that translating the materials from English to Dutch might have led to subtle differences in meaning. It is possible that in English, the term "drunk driving" is typically perceived as driving a vehicle when one is over the legal limit for drinking and driving, whereas the Dutch equivalent of "drunk driving" (i.e., "dronken rijden") is typically understood as driving a vehicle *when being drunk*. Given that the self-report questions asked about driving when being over the legal limit of drinking and driving and the category labels and items of the P-DUI-IAT included the term Dutch term for "drunk driving", it is possible that, in Study 1, the P-DUI-IAT only detected cases that were far over the legal limit for drinking and driving.

Study 1 also explored whether P-DUI-IAT scores prospectively predicted drunk driving over six months. Differences in IAT scores between the two groups were in the expected direction but were not statistically significant. This could be explained, however, by a lack of power to detect significant effects. Indeed, the sample sizes ($n = 17$ for the prospective drunk driving

group and *n* = 124 for the non-prospective drunk driving group) only allowed for 61% power to detect medium effect sizes in a between-groups comparison (*d* = .50, alpha = .05, one-tailed). Moreover, the Bayes factor showed only moderate evidence for the absence of the effect.

When using sample sizes that allowed higher statistical power to detect effects, results showed evidence for the utility of implicit measures of beliefs to prospectively predict drunk driving (in online samples). Results from Study 2 showed that both the P-DUI-IAT and A-DUI-IAT discriminated between participants who had driven drunk during the one-month follow-up period and participants who did not. Whereas results showed evidence for the utility of both IATs to independently predict prospective drunk driving, results did not show strong evidence for their utility to incrementally predict this outcome (the effect for incremental validity of the A-DUI-IAT was marginally significant: *p* = .048).

Finally, results from Study 2 provided evidence for the replicability of findings in previous studies [25] and Study 1. P-DUI-IAT scores were strongly related to drunk driving in the past year, drunk driving in the past month, and self-rated future likelihood of drunk driving. As opposed to results from Study 1, but in line with previous findings, results from Study 2 showed that P-DUI-IAT scores predicted past drunk driving outcomes above and beyond known risk factors. As opposed to results from previous studies [25], the current results did not show evidence for the utility of the P-DUI-IAT to incrementally predict self-rated future likelihood of drunk driving.

## Implications

Over the past 25 years, many studies have examined the predictive utility of implicit measures for several behavioral outcomes. Nevertheless, to this day, implicit measures are not applied in real-world contexts to predict behavior [30]. Potential reasons for this are that (a) traditional implicit measures typically show low predictive validity and (b) the practical utility of implicit measures is hardly being tested, or in other words, that research is not conducted for the purpose of bringing implicit measures into the real world. The current studies were designed while keeping in mind (a) recent developments in the field (i.e., using implicit measures of beliefs instead of using traditional implicit measures), (b) specific contexts in which the implicit measure of interest could have applied value, and (c) aspects that should be examined to assess practical utility (i.e., examining the predictive utility of implicit measures in a population for which they could have applied value and examine prospective predictive utility). Of course, the current studies only provide *initial* evidence for the practical utility of implicit measures and further research on other utility aspects will be necessary before the A- and P-DUI-IAT can be incorporated in real-world settings. Nevertheless, we believe that the findings from the current studies provide a first step towards that direction.

Both IATs could eventually be used in driving schools to predict which individuals are likely to drink and drive. Subsequently, those individuals could be provided with intervention measures (such as extra education) to prevent them from drinking and driving (again). The P-DUI-IAT could be used during the obligated refresher course to predict recidivism of drunk driving, while the A-DUI-IAT could be used to predict drunk driving in students who have not obtained their driver's license yet. While results from Study 2 did not show (strong) statistical evidence for incremental validity of our IATs in the prediction of prospective drunk driving, using the IAT in combination with other measures could be advantageous because the IAT is less susceptible to social desirability responding than other (self-report) measures.

## Limitations and future research

The current studies are not without limitations. First, translating the materials from English to Dutch might have led to subtle differences in meaning which could have resulted in difference

in findings between the study including Dutch-speaking participants and the studies including English-speaking participants. Future studies should take additional precautions before using the materials in different populations, such as conducting an analysis of conceptual equivalence (e.g., by consulting experts) and pilot testing the materials [31].

Second, we used self-reports as a criterion to test the validity of our IATs. As such, because of social desirability, some participants might not have truthfully reported their drunk driving behavior. Participants from online samples were probably more honest in reporting drunk driving behavior than participants from the ecologically valid sample (i.e., driving school students) because participants from the latter group were probably less inclined to put trust in our guarantees of anonymity (for example, because the invitation to participate in the study was sent out by driving schools). However, as discussed in the introduction of this paper, using self-reports to measure drunk driving behavior would be much more problematic in non-anonymous real-world settings where potential negative consequences (e.g., obligated training) are at stake. As such, implicit measures could have added value in applied contexts. Nevertheless, it remains difficult to demonstrate validity of implicit measures using self-reports as a criterion of drunk driving in ecologically valid contexts because even in an anonymous research context, these reports are probably less truthful. Future research should further validate the P- and A-DUI-IAT in ecologically valid contexts using more objective measures of drunk driving as a criterion (such as driving records).

Third, Study 1 had weak statistical power to test whether the P-DUI-IAT was able to prospectively predict drunk driving in driving school students. Future studies should systematically examine this question using well-powered study designs. Relatedly, future studies should test the predictive utility of the A-DUI-IAT in driving school samples. For practical application purposes, it would also be important for future studies to test whether the A-DUI-IAT can predict the onset of drunk driving behavior in such samples (note that we did not examine this in the current studies because they were not designed for this purpose). Third, to examine prospective predictive utility of our IATs, we used a relatively short follow-up period (i.e., one month). Future studies could examine prospective predictive utility of the IATs using longer follow-up periods (although for practical purposes it may be more valuable to know which individuals are at short-term risk).

Finally, while the IATs discriminated between participants who had driven drunk between baseline and follow-up and participants who did not, the classification statistics (as assessed through ROC analyses) were far from perfect. For our IATs to have practical value, these classification statistics should be improved and other classification statistics (e.g., positive predictive value) should be tested. To this end, future studies could tweak different aspects of the IATs (e.g., number of trials, category labels, etc.) and examine whether this improves their classification abilities. Also, before these measures can be applied in real-world contexts, their (other) psychometric properties should be examined within that specific context to ensure that the measures are valid and reliable.

## Conclusions

Results from the current studies showed initial evidence for the practical utility of implicit measures of beliefs for predicting drunk driving. More specifically, they showed evidence for (a) predictive utility of the P-DUI-IAT for drunk driving in driving school students, a sample for which this measure could have applied utility and (b) the utility of the A- and P-DUI-IAT to prospectively predict drunk driving. While further applied research is necessary, the current results could provide a first step towards the application of implicit measures in real-world contexts.

## Supporting information

**S1 Appendix. Exploratory analyses.**
(DOCX)

**S1 Table. Category labels and items for the past driving under the influence implicit association test.**
(DOCX)

**S2 Table. Category labels and items for the acceptability driving under the influence implicit association test.**
(DOCX)

**S3 Table. Risk factors for drunk driving outcomes (Study 1).**
(DOCX)

**S4 Table. Risk factors for drunk driving outcomes (Study 2).**
(DOCX)

## Acknowledgments

We would like to thank the driving schools that were involved in data collection.

## Author Contributions

**Conceptualization:** Femke Cathelyn, Pieter Van Dessel, Jan De Houwer.

**Methodology:** Femke Cathelyn, Pieter Van Dessel, Jan De Houwer.

**Writing – original draft:** Femke Cathelyn.

**Writing – review & editing:** Pieter Van Dessel, Jan De Houwer.

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
