## [Decision Letter · Decision Letter 0]

5 Aug 2022

PONE-D-21-38711Testing the practical utility of implicit measures of beliefs for predicting drunk drivingPLOS ONE

Dear Dr. Cathelyn,

Thank you for submitting your manuscript to PLOS ONE. After careful consideration, we feel that it has merit but does not fully meet PLOS ONE’s publication criteria as it currently stands. Therefore, we invite you to submit a revised version of the manuscript that addresses the points raised during the review process.

This decision letter replaces the previous decision on this manuscript. Please disregard any attachments to this message.

We look forward to receiving your revised manuscript.

Kind regards,

George Vousden

Deputy Editor in Chief

PLOS ONE

*On behalf of,*

Sónia Brito-Costa, Ph.D.

Academic Editor

PLOS ONE

Journal Requirements:

"This work is supported by Ghent University [grant number BOF16/MET_V/002; https://www.ugent.be/en/research/funding/bof/methusalem] to JDH  and by the Scientific Research Foundation Flanders [grant number FWO19/PDS/041; https://www.fwo.be/] to PVD. The funders had no role in study design, data collection and analysis, decision to publish, or preparation of the manuscript."

Reviewers' comments:

Reviewer's Responses to Questions

**Comments to the Author**

1. Is the manuscript technically sound, and do the data support the conclusions?

Reviewer #1: Yes

Reviewer #2: Partly

2. Has the statistical analysis been performed appropriately and rigorously? 

Reviewer #1: No

Reviewer #2: N/A

3. Have the authors made all data underlying the findings in their manuscript fully available?

Reviewer #1: Yes

Reviewer #2: Yes

4. Is the manuscript presented in an intelligible fashion and written in standard English?

Reviewer #1: Yes

Reviewer #2: Yes

5. Review Comments to the Author

Reviewer #1: Comments on PONE-D-21-38711

‘Testing the practical utility of implicit measures of beliefs for predicting drunk driving’

The topic of research is relevant for practical use in daily work and is an understudied subject. The authors are prominently working on ways to expand the scope and utility of the tool, which will ultimately help in protecting the lives of people. With few clarifications required in methodology, use of tools, and replicability of this study in other contexts, the proposed manuscript can be accepted for publication.

Few comments to adjust:

Abstract:

Always provide Confidence Internal and p-value while reporting OR results. Kindly include them.

For study 1: The limitations of not allowing to carry out confirmatory analysis with lower power and smaller sample size than anticipated (during the design) of the study are well explained. Thus, the preliminary concept of validation of predictive capacity of the tool in Dutch-speaking Belgian participants.

It is not clear which version of P-DUI-IAT – English or Dutch – was used. The previous study of the same group of researchers (Cathelyn, Van Dessel, & De Houwer, 2021), which was published in the Journal of Safety Research in October 2021, mentions that P-DUI-IAT was a newly developed tool by the researchers themselves. Not many psychometric properties except split-half reliability were presented. No confirmatory factor analysis was done. Similarly, the response rate in the follow-up study was not significant.

In any replication studies, the researchers have to adapt from failures and apply concrete procedures to overcome past mistakes. In study 2, the researchers replicated similar procedures of research with the inclusion of Native English-Speaking participants. The face validity, a roundtable with experts, pilot testing, and analysis of descriptive statistics before applying the full-scale study were undermined.

In study 2: the researchers changed the inclusion criteria and expanded to the participants from native English-Speaking countries. There is no demographic explanation on how many portions of the respondents were native English speakers and how many were non-native.

For me, the main issue of this study's findings is the replicability of the tool in the practical world. The content, face, concurrent, factorial, and external validity are still questionable. The tool still lacks a confirmatory factor analysis (even, exploratory one), and testing of other parameters (Souza, Alexandre, & Guirardello, 2017; Echevarría-Guanilo, Gonçalves, & Romanoski, 2018).

It is recommended to incorporate these elements, wherever feasible so that the findings of this study can easily be replicated, and the tool can be used properly.

Reference:

Cathelyn, F., Van Dessel, P., & De Houwer, J. (2021). Predicting Drunk Driving Using a Variant of the Implicit Association Test. Journal of Safety Research. https://liplab.be/onewebmedia/Cathelyn%20Van%20Dessel%20J%20Safety%20Res%20Drunk%20Driving%20Implicit.pdf

Souza, A. C. D., Alexandre, N. M. C., & Guirardello, E. D. B. (2017). Psychometric properties in instruments evaluation of reliability and validity. Epidemiologia e Serviços de Saúde, 26, 649-659.

Echevarría-Guanilo, M. E., Gonçalves, N., & Romanoski, P. J. (2018). Psychometric properties of measurement instruments: conceptual bases and evaluation methods-part I. Texto & Contexto-Enfermagem, 28: e20170311

Reviewer #2: I would like to congratulate the authors for the originality of the study. However, I think that it is possible to improve it.

I consider that the article is too long and it would be better to restructure it. For example, within the general structure of method, results and conclusions, include what pertains to each study, instead of each one of those sections for each study. It would also be appropriate to include more bibliography related to the main topic (for example, add more citations when referencing citation 22).

6. PLOS authors have the option to publish the peer review history of their article (what does this mean?). If published, this will include your full peer review and any attached files.

Reviewer #1: **Yes: **Yubaraj Adhikari, PhD

Reviewer #2: **Yes: **Marta Sancho

---

## [Author Response · Author response to Decision Letter 0]

9 Sep 2022

We thank the editor for inviting us to respond to the reviewers’ comments and revise the manuscript. We thank the reviewers for their constructive and valuable feedback. In the revised manuscript, we addressed the issues that were raised. Our specific responses to the reviewers’ comments are provided below. 

Reviewer #1

Comments on PONE-D-21-38711 ‘Testing the practical utility of implicit measures of beliefs for predicting drunk driving’. The topic of research is relevant for practical use in daily work and is an understudied subject. The authors are prominently working on ways to expand the scope and utility of the tool, which will ultimately help in protecting the lives of people. With few clarifications required in methodology, use of tools, and replicability of this study in other contexts, the proposed manuscript can be accepted for publication.

Reviewer 1 Comment #1 

Abstract: Always provide Confidence Internal and p-value while reporting OR results. Kindly include them.

Authors: We thank the reviewer for drawing our attention to this missing information. In the revised manuscript, we now always include confidence intervals and p-values for the ORs (e.g., Abstract, p.2, lines 35-41): 

“Results from Study 1 show that the P-DUI-IAT predicts self-rated past drunk driving behavior in driving school students (ORs = 3.11-6.12, ps < .043, 95% CIs = [1.11, 37.69]).” and “Results from Study 2, on the other hand, show strong evidence for the utility of both implicit measures to prospectively predict self-rated drunk driving (ORs = 3.80-5.82, ps < .002, 95% CIs = [1.72, 14.47]).” 

Reviewer 1 Comment #2

For study 1: The limitations of not allowing to carry out confirmatory analysis with lower power and smaller sample size than anticipated (during the design) of the study are well explained. Thus, the preliminary concept of validation of predictive capacity of the tool in Dutch-speaking Belgian participants. It is not clear which version of P-DUI-IAT – English or Dutch – was used. The previous study of the same group of researchers (Cathelyn, Van Dessel, & De Houwer, 2021), which was published in the Journal of Safety Research in October 2021, mentions that P-DUI-IAT was a newly developed tool by the researchers themselves. Not many psychometric properties except split-half reliability were presented. No confirmatory factor analysis was done. Similarly, the response rate in the follow-up study was not significant. 

Authors: We thank the reviewer for pointing out this ambiguity about the version of the IAT that we used. In the revised version of our manuscript, we now clearly state that “Five Belgian driving schools invited native Dutch-speaking students who had recently taken the refresher course to participate in Study 1.” (p.8, line 171) and “In Study 2, native English-speaking participants were recruited via Prolific Academic (an online recruitment platform).” (p.9, line 197). 

In the Materials section of the revised manuscript, we now also clearly state that “For Study 2, we adopted the (English) materials from our previous studies [25]. For Study 1, all materials were translated to Dutch using the back translation method.” (p.11, lines 227-228). 

We also agree with the reviewer that it is important to examine the psychometric properties of a measure before it can be applied in real-world contexts. Please note, however, that the aim of these studies was not to already deliver a measure that is ready to be used in a real-world context, and that examining all psychometric properties of our measures would go beyond the scope of this paper. Instead, the primary aim of the current studies was to test whether our measures can predict drunk driving (a) in an ecologically valid context and (b) over time. Thus, in the current studies, we examined the criterion/predictive validity (i.e., does our measure predict scores on a criterion measure?) and external/ecological validity (i.e., do our previous findings generalize to real-life situations outside of a research setting?). 

Also, while we did not examine the construct validity of our measures using a confirmatory (or exploratory) factor analysis, we did examine this type of validity using the known-groups technique. More specifically, by assessing whether our measures can detect differences between drunk driving and non-drunk driving groups (as described in Souza et al., 2017). Thus, we believe that these (and our previous) studies provide initial evidence for the validity of our measures, which is the aim of our studies. 

Of course, a more distal goal of these studies is to develop a measure that could eventually be applied in real-world contexts (e.g., driving schools) to predict drunk driving, and we agree that further investigation of the psychometric properties of our measures will be necessary before application can be considered viable. Therefore, in the revised version of the manuscript, we now note that “Before these measures can be applied in real-world contexts, their psychometric properties (e.g., test-retest reliability, divergent validity, etc.) should be examined within those specific contexts to ensure that the measures are valid and reliable.” (p.27, lines 603-605). 

Reviewer 1 Comment #3 

In any replication studies, the researchers have to adapt from failures and apply concrete procedures to overcome past mistakes. In study 2, the researchers replicated similar procedures of research with the inclusion of Native English-Speaking participants. The face validity, a roundtable with experts, pilot testing, and analysis of descriptive statistics before applying the full-scale study were undermined.

Authors: Please note that (as described in our response to Comment #2 by Reviewer 1) in Study 2, we adopted the original English materials from our previous studies (Cathelyn et al., 2021). For Study 1, however, we did translate the IAT materials to Dutch using the back translation method which might have led to differences in how IAT categories can be interpreted compared to in the original IAT. We agree that this is a limitation of Study 1. In the General Discussion of the revised manuscript, we now acknowledge this limitation: 

“Translating the materials from English to Dutch might have led to subtle differences in meaning which could have resulted in difference in findings between the study including Dutch-speaking participants and the studies including English-speaking participants. Future studies should take additional precautions before using the materials in different populations, such as conducting an analysis of conceptual equivalence (e.g., by consulting experts) and pilot testing the materials (Echevarría-Guanilo et al., 2018)” (p.26, lines 565-570)

Reviewer 1 Comment #4

In study 2: the researchers changed the inclusion criteria and expanded to the participants from native English-Speaking countries. There is no demographic explanation on how many portions of the respondents were native English speakers and how many were non-native. 

Authors: In Study 2, participants were recruited via Prolific Academic (an online recruitment platform). We used two filters to ensure that participants were native-English speakers (i.e., first language English and UK or USA nationality). This was already reported in the previous version of our manuscript: “Participants who owned a valid driver’s license, drove their car at least once per week, drank more than one unit of alcohol per week, had the UK nationality, and whose first language was English, were invited to participate in the prescreening study.” (p. 9, lines 200-203) 

Reviewer 1 Comment #5 

For me, the main issue of this study's findings is the replicability of the tool in the practical world. The content, face, concurrent, factorial, and external validity are still questionable. The tool still lacks a confirmatory factor analysis (even, exploratory one), and testing of other parameters (Souza, Alexandre, & Guirardello, 2017; Echevarría-Guanilo, Gonçalves, & Romanoski, 2018). It is recommended to incorporate these elements, wherever feasible so that the findings of this study can easily be replicated, and the tool can be used properly.

Authors: We agree that our measures are not ready to be applied in real-world contexts, and we explicitly comment on this in the General Discussion of our manuscript, for example: 

“Of course, the current studies only provide initial evidence for the practical utility of implicit measures and further research on other utility aspects will be necessary before the A- and P-DUI-IAT can be incorporated in real-world settings. Nevertheless, we believe that the findings from the current studies provide a first step towards that direction.” (p.25, lines 551-554). 

“Finally, while the IATs discriminated between participants who had driven drunk between baseline and follow-up and participants who did not, the classification statistics (as assessed through ROC analyses) were far from perfect. For our IATs to have practical value, these classification statistics should be improved and other classification statistics (e.g., positive predictive value) should be tested. To this end, future studies could tweak different aspects of the IATs (e.g., number of trials, category labels, etc.) and examine whether this improves their classification abilities. Also, before these measures can be applied in real-world contexts, their (other) psychometric properties should be examined within that specific context to ensure that the measures are valid and reliable.” (p.27, lines 597-605)

And, as noted in the Conclusions section of our manuscript: “While further applied research is necessary, the current results could provide a first step towards the application of implicit measures in real-world contexts.” (p.28, lines 611-613) 

Please also see our response to Comment #2 by Reviewer 1. 

Reviewer #2

I would like to congratulate the authors for the originality of the study. However, I think that it is possible to improve it.

Reviewer #2 Comment #1:

I consider that the article is too long and it would be better to restructure it. For example, within the general structure of method, results and conclusions, include what pertains to each study, instead of each one of those sections for each study. It would also be appropriate to include more bibliography related to the main topic (for example, add more citations when referencing citation 22).

Authors: We thank the reviewer for drawing our attention to this issue. We followed the reviewer’s suggestion to restructure the manuscript such that there is now one Method, Results, and Discussion section for both studies instead of individual subsections for each study. 

In the revised manuscript, we also discuss in more detail the literature related to the utility of implicit measures (see for example p.3, line 49, and p.4 line 72). Please note that we were not able to add more references regarding the prediction of road safety behavior using implicit measures because this is a relatively new field and we believe that we already discussed all the relevant literature. For example, to the best of our knowledge, only two studies thus far have tested the utility of (more traditional) implicit measures for predicting drunk driving. We also refer to a recent review by Tosi et al. (2021) on implicit and explicit measures in transportation research. Also, please note that citation 22 (now citation 25) refers to our previous study in which we developed an implicit measure of beliefs for predicting drunk driving. This was the first study to develop this type of measure for these purposes. The aim of the current studies was to build on this previous study and further validate the P-DUI-IAT as a tool for predicting drunk driving. 

References 

Cathelyn, F., Van Dessel, P., & De Houwer, J. (2022). Predicting drunk driving using a variant of the implicit association test. Journal of safety research, 81, 134-142.

Echevarría-Guanilo, M. E., Gonçalves, N., & Romanoski, P. J. (2018). Psychometric properties of measurement instruments: conceptual bases and evaluation methods-part I. Texto & Contexto-Enfermagem, 26.

Souza, A. C. D., Alexandre, N. M. C., & Guirardello, E. D. B. (2017). Psychometric properties in instruments evaluation of reliability and validity. Epidemiologia e servicos de saude, 26, 649-659.

Tosi, J.D., Haworth, N., Díaz-Lázaro, C.M., Poó, F.M., Ledesma, R.D. (2021). Implicit and explicit attitudes in transportation research: A literature review. Transportation Research Part F: Traffic Psychology and Behaviour, 77. 87–101.

---

## [Decision Letter · Decision Letter 1]

14 Sep 2022

Testing the practical utility of implicit measures of beliefs for predicting drunk driving

PONE-D-21-38711R1

Dear Dr. Cathelyn,

We’re pleased to inform you that your manuscript has been judged scientifically suitable for publication and will be formally accepted for publication once it meets all outstanding technical requirements.

Kind regards,

Sónia Brito-Costa, Ph.D.

Academic Editor

PLOS ONE

Additional Editor Comments (optional):

Reviewers' comments:

Reviewer's Responses to Questions

**Comments to the Author**

1. If the authors have adequately addressed your comments raised in a previous round of review and you feel that this manuscript is now acceptable for publication, you may indicate that here to bypass the “Comments to the Author” section, enter your conflict of interest statement in the “Confidential to Editor” section, and submit your "Accept" recommendation.

Reviewer #1: All comments have been addressed

2. Is the manuscript technically sound, and do the data support the conclusions?

Reviewer #1: Yes

3. Has the statistical analysis been performed appropriately and rigorously? 

Reviewer #1: I Don't Know

4. Have the authors made all data underlying the findings in their manuscript fully available?

Reviewer #1: Yes

5. Is the manuscript presented in an intelligible fashion and written in standard English?

Reviewer #1: Yes

6. Review Comments to the Author

Reviewer #1: The authors have adequately addressed all concerns and comments raised by the reviewer. The reviewer believes that the quality of the paper is improved after addressing the concerns. The reviewer recommends to accept this paper and publish accordingly.

7. PLOS authors have the option to publish the peer review history of their article (what does this mean?). If published, this will include your full peer review and any attached files.

Reviewer #1: **Yes: **Yubaraj Adhikari, PhD

---

## [Editor Report · Acceptance letter]

20 Sep 2022

PONE-D-21-38711R1 

Testing the practical utility of implicit measures of beliefs for predicting drunk driving 

Dear Dr. Cathelyn:

I'm pleased to inform you that your manuscript has been deemed suitable for publication in PLOS ONE. Congratulations! Your manuscript is now with our production department. 

Kind regards, 

on behalf of

Dr. Sónia Brito-Costa 

Academic Editor

PLOS ONE